# Physiological and stem cell compartmentalization within the Drosophila midgut

**Alexis Marianes, Allan C Spradling\***

Department of Embryology, Howard Hughes Medical Institute, Carnegie Institution for Science, Baltimore, United States

**Abstract** The Drosophila midgut is maintained throughout its length by superficially similar, multipotent intestinal stem cells that generate new enterocytes and enteroendocrine cells in response to tissue requirements. We found that the midgut shows striking regional differentiation along its anterior-posterior axis. At least ten distinct subregions differ in cell morphology, physiology and the expression of hundreds of genes with likely tissue functions. Stem cells also vary regionally in behavior and gene expression, suggesting that they contribute to midgut sub-specialization. Clonal analyses showed that stem cells generate progeny located outside their own subregion at only one of six borders tested, suggesting that midgut subregions resemble cellular compartments involved in tissue development. Tumors generated by disrupting Notch signaling arose preferentially in three subregions and tumor cells also appeared to respect regional borders. Thus, apparently similar intestinal stem cells differ regionally in cell production, gene expression and in the ability to spawn tumors.

## Introduction

Stem cells allow a tissue to maintain homeostasis by regularly replacing old and damaged cells (reviewed in *Losick et al., 2011*). For example, intestinal stem cells (ISCs) of the adult Drosophila midgut (*Ohlstein and Spradling, 2006*; *Micchelli and Perrimon, 2006*) generate the two major differentiated cell types of the gut, enterocytes and enteroendocrine cells, throughout life. ISCs with high levels of the Delta (Dl) ligand specify daughters to be enterocytes via unidirectional Notch signaling, while ISCs with low Delta levels specify enteroendocrine cell daughters (*Micchelli and Perrimon, 2006*; *Ohlstein and Spradling, 2006*, *2007*). Although, the exact nature of their niche remains unclear, ISCs are highly responsive to signals from neighboring cells. The animal's nutritional state, as sensed in part by insulin signaling, influences ISC division and whether divisions are symmetric or asymmetric (*Choi et al., 2011*; *O'Brien et al., 2011*). Dietary toxins, pathogenic bacteria, or other stressors stimulate enterocytes to release Egfr and JAK/STAT ligands that trigger nearby stem cells to multiply and produce tissue hyperplasia (reviewed in *Jiang and Edgar, 2011*).

ISCs that follow a generally similar program of enterocyte and enteroendocrine cell production based on Notch signaling were detected along the entire length of the midgut by clonal analysis and marker gene expression (*Ohlstein and Spradling, 2006*; *Micchelli and Perrimon, 2006*; *Ohlstein and Spradling, 2007*). Whether these stem cells are precisely equivalent, however, has not been critically tested since most studies have focused on the posterior midgut, where ISC divisions are frequent. Some differences are plausible, since histological studies have long distinguished the middle midgut from the anterior and posterior segments. The middle midgut has an acidic pH, and is itself tripartite in organization. The anterior portion contains specialized enterocytes known as interstitial cells (*Filshie et al., 1971*), interspersed with morphologically distinctive 'copper cells', which selectively fluoresce following exposure to copper ions (*Poulson and Bowen, 1952*; *McNulty et al., 2001*). There follows a zone of 'large flat cells' (*Poulson and Waterhouse, 1960*) and an 'iron region' enriched in the iron-storage

\*For correspondence: spradling@ciwemb.edu

**Competing interests:** The authors declare that no competing interests exist.

**Reviewing editor**: Andrea Brand, University of Cambridge, United Kingdom

**eLife digest** Many cells in the body accumulate wear and tear over time, and a fraction of them are always nearing the end of their lives. However, in some tissues there are stem cells that can divide into daughter cells which then differentiate and replace the damaged cells. Unlike embryonic stem cells, these 'adult tissue stem cells' normally differentiate into only a few related cell types, but their ability to produce replacement cells keeps the tissue functioning normally. Here, Marianes and Spradling have investigated a type of adult stem cell, known as intestinal stem cells, that resides in the midgut of fruit flies.

The midgut is the major site of digestion in fruit flies, and functions much like the small intestine in mammals. This tissue is a long tube that is lined with two types of cells: digestive cells and hormone-producing cells. These cell types are maintained by thousands of apparently similar intestinal stem cells, and it has long been thought that the stem cells give rise to cells throughout the midgut by responding to the same set of signals. However, certain digestive processes—such as the breakdown or uptake of particular nutrients—are known to occur only in a specific portion of the intestine. For example, in fruit flies, a region in the middle of the intestine is acidified, and may act like an extra stomach. And in both fruit flies and mammals, iron is taken up mostly in the area of the gut just after the stomach. These regional differences in function have led to uncertainty over how midgut cells both arise and are replaced.

Marianes and Spradling now show, based on a detailed study of tissue cells and stem cells, that the midgut contains at least ten subregions that occur in a specific order. The cells in these subregions have distinct features, including shape, size and contents (e.g., stores of carbohydrates or nutrients). Each subregion appears to perform specific functions during digestion, and the cells in these subregions also transcribe genes that reflect their roles in breaking down or storing various nutrients. Interestingly, the stem cells in most subregions are distinct, and do not differentiate into the cells from adjacent subregions. The subregions also differ in their incidence of cancer: when a particular signal was inhibited in stem cells in all ten subregions, aggressive tumors formed in only three subregions and the tumor cells did not cross into neighboring subregions.

These observations may inform future studies of the mammalian small intestine and improve our understanding of its susceptibility to cancer and other diseases.

protein Ferritin (*Poulson and Waterhouse, 1960*; *Poulson and Bowen, 1952*; *Mehta et al., 2009*). Copper cells depend uniquely on the homeotic gene *labial* during embryonic differentiation (*Panganiban et al., 1990*; *Hoppler and Bienz, 1994*; *Dubreuil et al., 2001*) and function in acid production using vacuolar $H^+$ ATPase pump proteins localized on their apical membranes (*Dubreuil, 2004*; *Shanbhag and Tripathi, 2009*). A recent study of ISCs in the copper region (*Strand and Micchelli, 2011*) found that they are capable, like posterior ISCs, of replenishing all the major cell types, including copper, interstitial and enteroendocrine cells. However, copper region ISCs were reported to differ from posterior ISCs in lacking the Notch ligand Delta, and in being normally quiescent in the absence of stress (*Strand and Micchelli, 2011*). Thus, the regulation of ISCs differs in the copper region compared to other studied regions of the midgut.

The possible existence of regional variation is further suggested by the restricted spatial localization of some digestive enzymes in midguts from a variety of insects (reviewed by *Terra and Ferreira, 1994*) and from Drosophila larvae. Some enzymes, such as the lipase Magro (*Sieber and Thummel, 2012*), may be trafficked into the midgut from the proventriculus via the peritrophic matrix (*King, 1988*). Others such as, α-amylase, which is expressed primarily in the anterior and posterior midgut regions (*Thompson et al., 1992*) probably indicate true regional differences in enterocyte expression. Some of the strongest evidence for further regionalization comes from studies showing that unique neuropeptides are secreted by enteroendocrine cells located in specific gut regions (*Ohlstein and Spradling, 2006*; *Veenstra et al., 2008*). These spatial differences in gene expression might be induced downstream of the ISC by region-specific signals, or they might reflect intrinsic differences in regional stem cell programming.

Here we document extensive regionalization along the length of the Drosophila midgut, at the level of morphology, cell behavior and gene expression. Each subregion displays a sharp boundary with its

neighbors, suggesting that it carries out distinctive functions. ISCs likely contribute to these differences, since stem cells from most tested regions did not produce adjacent region cells even when located at the border. Regional stem cell differences likely influence tumorigenesis, since midgut tumors caused by attenuating Notch signaling arose at very different rates in the different subregions. Thus, tissue stem cells may comprise a wider variety of types, each with a more limited therapeutic scope, than previously appreciated.

## Results

### Regionalized morphological ultrastructure along the midgut a/p axis

The Drosophila intestine varies significantly in cellular content and activity based on age, sex, mating status, and nutritional and environmental conditions (*Ohlstein and Spradling, 2006*; *O'Brien et al., 2011*; reviewed in *Jiang and Edgar, 2011*). We used stringent animal husbandry strategies to minimize such variation. Only, fertilized adult females 4–14 days of age were employed to avoid the final steps of gut maturation that take place in young adults (*Takashima et al., 2013a*), as well as age-induced decline (*O'Brian et al., 2011*). Flies were kept at a controlled density in fresh vials, at 25°C and provided with a uniform level of nutrition before and during the study period. Under these conditions, the cellular structure of the midgut was stable and reproducible as assessed by cell counts along its length (*Figure 1*). Our rationale was to understand a 'steady state' gut before analyzing the more complex situations where the gut is changing its structure (*O'Brien et al., 2011*).

Under these conditions, we looked for differences in enterocyte morphology along the a/p axis of the Drosophila midgut using light and electron microscopy (EM). EM analyses of longitudinally sectioned midguts revealed many more reproducible regional differences in enterocyte structure than those described previously in the middle midgut, including differences in cell type, cell size, membrane invaginations, microvillar length, and the presence of ferritin particles, glycogen or lipid droplets (*Figure 1—figure supplement 1*). Differences were summarized in scale drawings that define 10 distinct zones along the a/p axis (*Figure 1A*). Furthermore we used nutrient-specific and antibody stains, as well as protein trap lines to validate many of the subregions. Ferritin-GFP selectively stained the Fe region, while Nile red, which stains lipid droplets, marked the A2, P1 and P3 zones (*Figure 1B*), the same regions showing enterocytes with high lipid content in EM images. The number of enterocyte cell diameters along the a/p axis within each section was recorded for multiple guts and these proved to be highly reproducible (*Figure 1C*).

### GAL4 expression patterns correspond to midgut structural features

We searched for gene expression differences that would mirror these variations in enterocyte structure. Since GAL4 driver lines would be particularly useful for subsequent functional tests, we screened the midgut expression patterns of 931 Gal4 lines from the Janelia Farm collection (*Jenett et al., 2012*) and eight additional GAL4 strains. The GFP expression patterns driven by the most useful lines in 5–10 day old adult female midguts (*Figure 1—figure supplement 2*) are summarized in *Figure 1D–E*. One example (*Figure 1F*) shows expression only in A1, while another labels the Cu region, as well as in P2-4 (*Figure 1G*). By counting the number of enterocytes along the a/p axis, bands of GFP expression were approximately mapped to subregions (*Figure 1F–G*, dashed lines) in each particular cell type (*Figure 1—figure supplement 2*). The correspondence of gene expression boundaries in endodermal cells with the morphological junctions of the 10 subregions was striking (*Figure 1D*). In contrast, gene expression boundaries in gut muscle were mostly offset from these boundaries (*Figure 1E*).

### Gene expression varies extensively between midgut subregions

We next investigated whether physiologically meaningful differences in endogenous gene expression occur in the different regions of the midgut. For this purpose, we used appropriate GAL4 lines to drive UAS-GFP fluorescence, and manually isolated specific gut subregions under a fluorescence microscope. Dissected regions were rapidly isolated in small quantities, and total cellular RNA was extracted using a protocol that prevented RNA degradation. RNA samples from 30 separate preparations derived from 10 different single or grouped regions (*Figure 2A*) were analyzed by RNAseq using paired-end reads of 100 × 100 nucleotides and a depth of 30–110 million reads (*Supplementary file 1A*).

The sequence data confirmed that specific regions had been isolated free of significant cross-contamination and revealed a much greater level of regionalized gene expression than previously

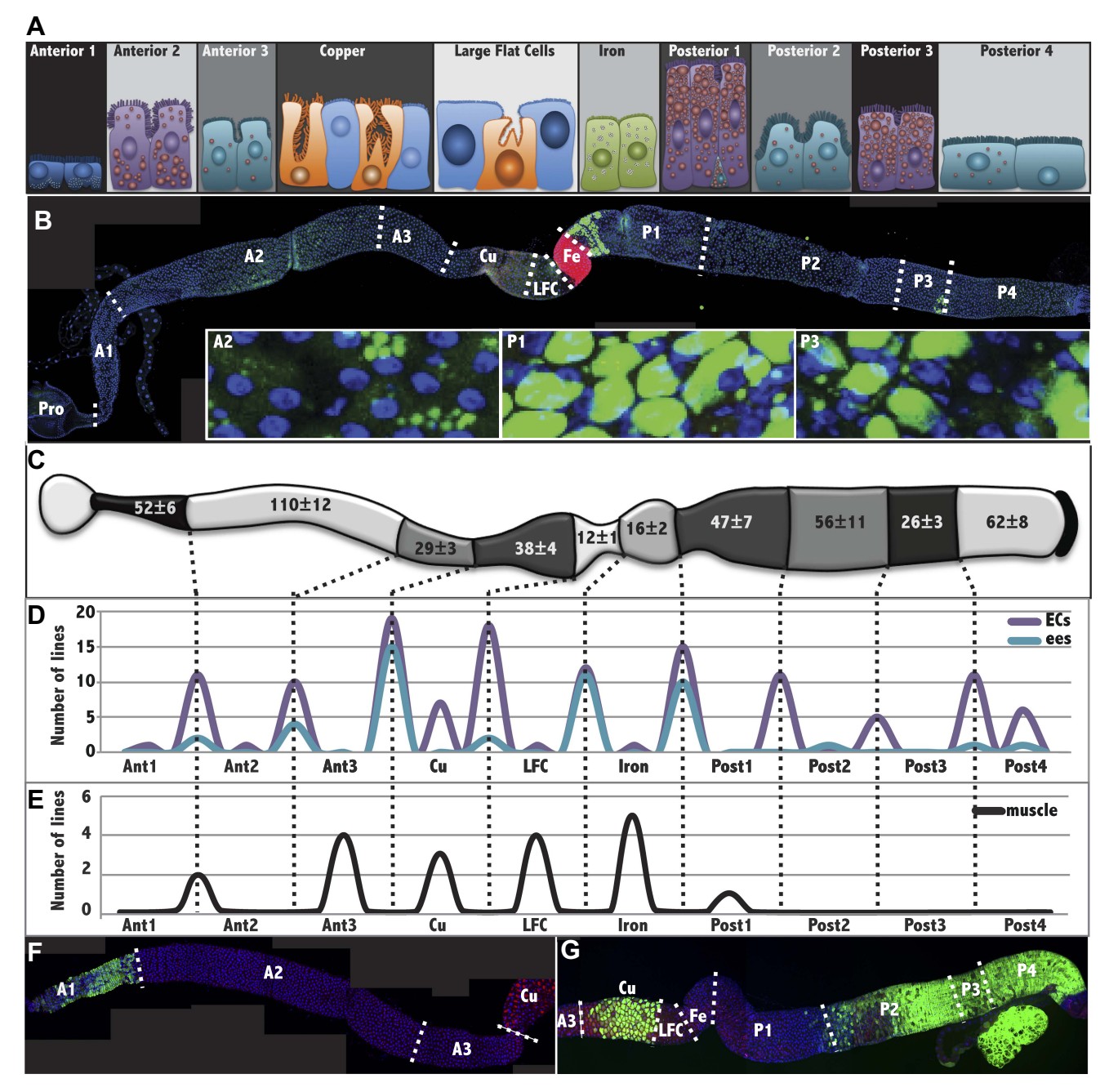

Figure 1. The Drosophila midgut is comprised of multiple subregions. (A) Drawings are shown to scale of representative enterocytes from each of 10 consecutive regions located along the anterior-posterior (a/p) axis of the midgut, as determined from electron microscopy, including parietal cells (orange) and interstitial cells (sky blue). Cells enriched in glycogen (navy blue), lipid (purple or teal), or iron (green) are shown. (B) A midgut from a Fer-GFP protein trap stained anti-GFP (red) to highlight the Fe region, and with Nile red (green) to highlight lipid-rich regions A2, P1, and P3. Inset: lipid staining in the indicated zones is shown at higher magnification. (C) Midgut drawing showing the average number of enterocyte cell diameters along the a/p axis within the 10 subregions (± SEM). (D) Gene expression borders in enterocytes (ECs, purple) or enteroendocrine cells (ees, blue) at the indicated subregion junctions are totaled for 49 Janelia lines with patterned midgut expression. (E) Gene expression borders in midgut–associated muscle relative to subregion junctions are totaled for the eight Janelia lines with patterned circular muscle expression. (F and G) Midguts from a R45D10-GAL4; UAS-GFP female (F) or R50A12-GAL4; UAS-GFP female (G) stained with anti-Cut (red polyploid Cu cells), anti-Prospero (red diploid cells throughout), and anti-GFP antibody shows expression restricted to A1 or to Cu, P2, P3 and P4, respectively.

Figure 1. Continued on next page

*Figure 1. Continued*

The following figure supplements are available for figure 1:

**Figure supplement 1**. Electron microscopic analysis of midgut enterocytes reveals ten subregions along the anterior-posterior axis (see diagram above).

**Figure supplement 2**. Gene expression patterns of Janelia and other useful Gal4 lines.

supposed. For example, the expression of the neuropeptides Npf and Ast-A is patterned within the midgut (*Veenstra et al., 2008*) and we observed expression of the corresponding mRNAs within the middle and posterior regions, as predicted, along with patterned expression of many other previously unlocalized neuropeptide mRNAs (*Figure 2B,D*). The *labial (lab)* gene encodes a homeobox transcription factor that is only expressed in the copper cell region, where it is required for copper cell differentiation (*Panganiban et al., 1990*; *Hoppler and Bienz, 1994*; *Dubreuil et al., 2001*; *Strand and Micchelli, 2011*). The RNAseq data showed that labial reads were detected only within the middle midgut in experiments that divided the midgut into anterior (A1, A2 and A3), middle (Cu, LFC and Fe) or posterior (P1, P2, P3 and P4) sections. Revealingly, *lab* fpkm was 40–100 times higher in isolated Cu region RNA than in any other adult midgut region (*Figure 2D*). All seven of the midgut segments we were able to isolate showed highly specific gene expression. CG10725 was enriched in A1 (*Figure 2D*), along with many other genes involved in chitin metabolism, where it likely encodes a component of the peritrophic membrane. Although their levels of spatial specificity were not previously known, Beta-galactosidase (Gal) mRNA was selectively found in the A2-A3 sample, the ZIP1 iron transporter mRNA in the Fe region, and the CG3106 acyl transferase mRNA in P1. This high specificity proves that cross-contamination was minimal and that gene expression boundaries correspond to the regional boundaries used in tissue isolation, while the correspondence of the RNAseq data to previous studies of gene expression validates the reliability of this large new body of information.

Many genes were expressed in previously unappreciated regional patterns, suggesting they are relevant to diverse aspects of midgut function. For example, PGRPs are membrane proteins that mediate interactions between the midgut and the microbiome that are essential for normal digestion (*Cash et al., 2006*; *Ryu et al., 2008*). *PGRP-SC1b* and *PGRP-SC2* expression was found to be highly enriched in the LFC and Fe regions, respectively, along with their regulator, *caudal* (*Figure 2E*). The localized expression of these and other PGRP family members (*Figure 2C*) suggest that the microbiome interacts with enterocytes in a highly organized way and that some bacterial species may be localized to specific regions of the gut lumen by the selective expression of host factors.

Regionally expressed genes were surprisingly common and specific. Each isolated subregion expressed 50–150 genes 10 times higher than in any other selected subregion of the midgut (*Supplementary file 1B*). The anterior, middle and posterior zones each expressed 259–474 genes at least five times more highly than in the other two zones, and differentially expressed mRNAs represented some of the most highly expressed genes in each region (*Supplementary file 1B*). Many differentially expressed genes were annotated as enzymes likely to be relevant to digestion or metabolism, or their expression had previously been mapped specifically to larval or adult midgut by the Fly Atlas microarray project (*Chintapalli et al., 2007*). Interestingly, many regionally expressed midgut genes reside in genomic clusters like highly expressed genes in many other differentiated cells (*Spellman and Rubin, 2002*). These included trypsins, lysozymes, serine proteases, alpha-glucosidases, lipases and Jonah genes (*Akam and Carlson, 1985*).

We carried out gene ontology analyses of the differentially expressed genes and abundant genes from each region and identified candidate biochemical pathways to which they likely contribute using the Database for Annotation, Visualization, and Integrated Discovery (DAVID) (*Huang et al., 2008a*, *2008b*). *Figure 3* shows GO terms enriched in the anterior, middle or posterior midgut, as well as in each of the individual regions that were isolated. These findings suggest several general insights into midgut function. Food entering the anterior midgut from the proventriculus begins to break down due to the action of enzymes expressed in the anterior midgut. In addition to digestive enzymes, anterior enterocytes may secrete other substances such as lipids to aid in nutrient solubilization. Antibacterial activity is also important in this region. In the three middle midgut regions food is further digested releasing sugars, amino acids, fatty acids, etc aided by the low pH. Additionally, metal ions that must be reduced before they can be absorbed, and other micronutrients are likely taken up for storage or

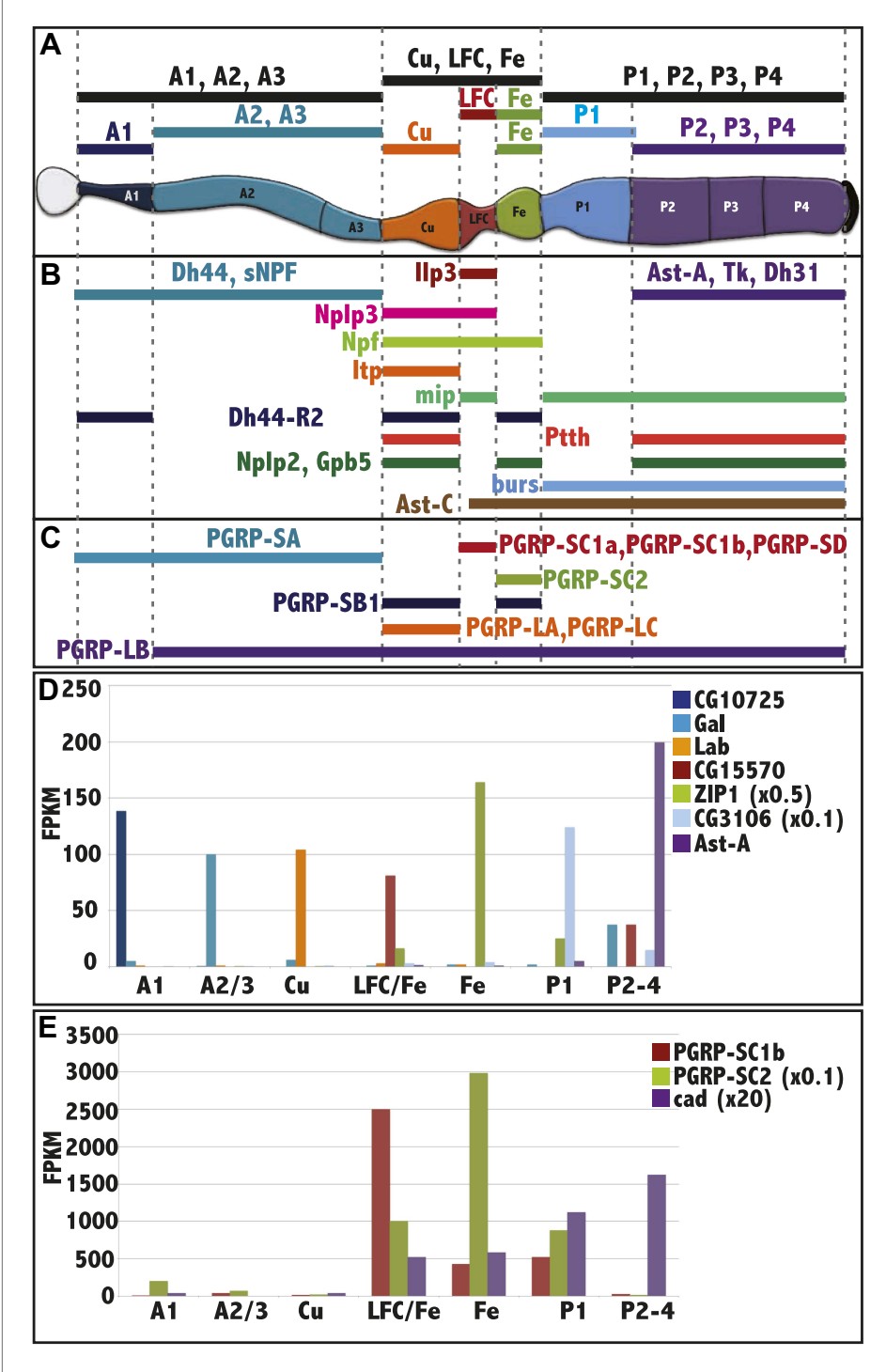

**Figure 2**. RNAseq analysis of midgut subregions. (**A**) Schematic of the 10 samples collected. Bars represent isolated portions of gut tissue, either containing single regions (A1, Cu, Fe, P1) or pooled regions (A1-3, Cu-LFC-Fe, P1-4, A2-3, LFC-Fe, P2-4). (**B**) Expression profile of select hormones showing regional specificity. (**C**) Expression profile of PGRP isoforms showing regional specificity. (**D**) Regionalized gene expression validates RNAseq method and rules out cross contamination. For each gene (key at right), its mean expression level (fpkm) by RNAseq in the three replicate isolated regions (x axis) is plotted. CG10725 (peritrophic membrane), Gal (beta-Galactosidase), lab (homeobox transcription factor), CG15570 (unknown), ZIP1 x 0.5 (zinc-iron transporter), CG3106 x 0.1 (acyl transferase), Ast (Allatostatin). (**E**) Regionalized expression of *caudal (cad)* and two PGRP genes implicated in immune interactions with the midgut microbiome.

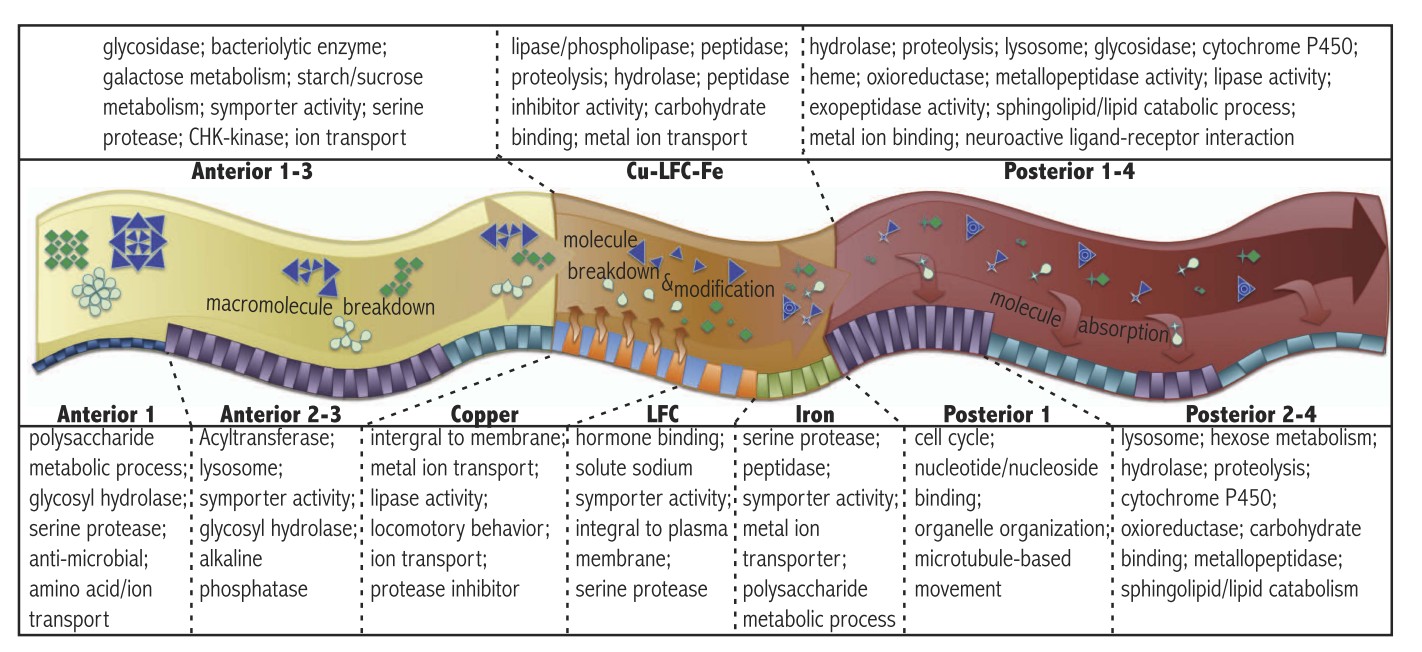

**Figure 3**. Regional gene expression. Pathways and gene ontology analysis using DAVID (see 'Materials and methods') enriched in the anterior (A1–3 pooled sample), middle (Cu-LFC-Fe pooled sample), and posterior (P1–4 pooled sample) midgut are listed in the upper panel. Below, a schematic diagram illustrates sequential strategy of digestion suggested by RNAseq analysis. Orange arrows from orange parietal cells in copper region indicate acidification process. Bottom panel indicates pathways and gene ontology analysis using DAVID enriched in subregions.

metabolism. Finally, upon entry into the posterior midgut, nutrients begin to be massively absorbed by enterocytes and transported to other regions of the body for storage, modification or utilization. The lipid-rich zone in the P1 region probably represents such nutrient uptake. The large flux of ready-to-use nutrients in the posterior region may explain why their stem cells divide more rapidly than other regions (*Figure 4S*; *Ohlstein and Spradling, 2006*; *Micchelli and Perrimon, 2006*). Both the middle and posterior regions likely maintain large populations of commensal microorganisms to aid in carrying out their functions.

## Stem cells differ between some subregions

We analyzed the ISCs within all 10 subregions to look for morphological and gene expression differences that might contribute to regional specialization. ISCs can be recognized in electron micrographs due to their basal location, extensive basement membrane contact, and triangular shape (*Figure 4A*). In A1, where enterocytes are more squamous and shorter than in other regions, the stem cells were flatter than the upward-directed triangular shape seen in all other regions (*Figure 4B*). Additionally, in the most lipid-rich region, P1, ISCs often contained many lipid droplets and reported lipid synthesis using the SREBP-GAL4 reporter line (*Matthews et al., 2009*) (*Figure 4C*). We verified that these cells were ISCs by staining for Delta and with Nile red (*Figure 4D*). Lipid droplets were also observed in A2 ISCs but in much fewer numbers (not shown). Two lines uncovered in the screen, R42G03 (Pdp1) and R44E05 (Stat92E), were differentially expressed in ISCs (*Figure 4E*; *Figure 1—figure supplement 2*), and the boundaries of expression coincided with regional boundaries. Thus, stem cells can be regionally distinctive, but it remained unclear if these differences were a cause or effect of other regional features.

Studies of stem cell behavior throughout the midgut uncovered further regional differences. The frequency of ISCs was investigated by staining for Delta (*Figure 4F–O*). Using improved staining conditions ('Materials and methods'), we detected Dl+ cells with the characteristics of ISCs in all 10 regions (*Figure 4F–O*), including the Cu region (*Figure 4I*), where Delta expression in ISCs was previously reported to be absent (*Strand and Micchelli, 2011*). However, ISCs divided at different rates in different regions based on clonal marking (*Figure 4S*). While it has been reported that heat shock can alter ISC division rates measured by lineage labeling (*Strand and Micchelli, 2011*),

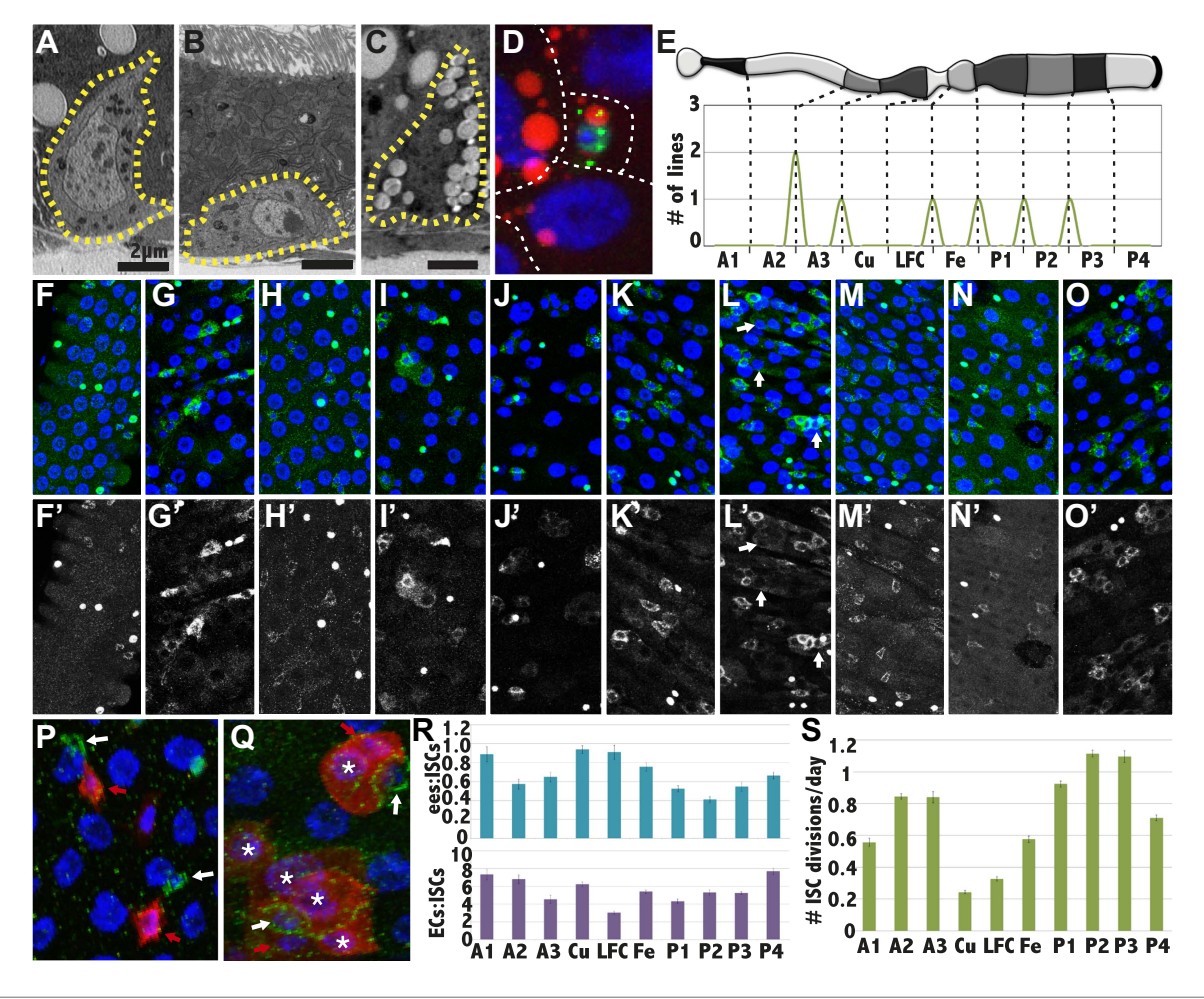

**Figure 4**. Stem cells also differ between subregions. (**A–C**) Electron micrographs of an intestinal stem cell from regions: A3—representing typical stem cell morphology—(**A**), A1 showing its flatter shape (**B**), P1 showing the presence of lipid droplets. (**D**) Light micrograph of a P1 ISC stained with anti-Delta (green), Nile red (red) and DAPI (blue). (**E**) Expression boundaries in two GAL4 lines with ISCs expression. (**F–O**) Light micrograph of ISCs (green cytoplasm), ees (green nuclear), and enterocytes (polyploid DAPI nuclei) displaying regional cell ratios. (**F**) A1, (**G**) A2, (**H**) A3, (**I**) Cu, (**J**) LFC, (**K**) Fe, (**L**) P1, (**M**) P2, (**N**) P3, (**O**) P4, (**P–Q**) Light micrograph of midgut regions stained for Delta (green) and for Notch reception activity as assayed by GbeSu(H)-LacZ (red). White arrow: Dl+ ISC. Red arrow: GbeSu(H)-LacZ+ EB. White asterisk: GbeSu(H)-LacZ+ EC. (**P**) A2 region showing recently decided call pairs with Delta staining (green) confined to ISCs (white arrows) and Notch reporter activity (red) in EBs. (**Q**) P1 region stained as in (**P**), showing that Notch activation (red) and Delta expression (green) persists downstream from the EB into young ECs. (**R**) Ratio of enteroendocrine cells to ISCs (blue, top) and ratio of enterocytes to ISCs (purple, bottom) in different midgut regions. (**S**) ISC divisions per day in different midgut regions as determined from clonal analysis.

we minimized these effects by using a single brief heat shock and a marking system that lacks GAL4. Moreover, all subregions should have been affected equally, and the data should be accurate as relative measurements. The most rapidly dividing ISCs are found in P1-P3, which divide about once per day, while ISC division is slightly less frequent in the anterior midgut, and much slower in the middle region. However, in contrast to a previous report (*Strand and Micchelli, 2011*), Cu region ISCs were not entirely quiescent but divided regularly every 4–5 days. ISCs in different regions maintain differing numbers of enteroendocrine cells and enterocytes (*Figure 4R*). Interestingly, in P1 (and to a lesser extent in P3 and A2), the enteroblast and even the youngest EC continued to stain for Dl (*Figure 4L'*). Although ISCs generated the same cells as elsewhere in the midgut, the persistent Delta in enteroblasts in the P1 region likely leads to persistent Notch activation during early enterocyte differentiation; the Notch activity reporter GbeSu(H)-lacZ was expressed in young ECs in P1 and P3, but not in other regions (*Figure 4Q vs 4P*).

## Most stem cells generate progeny only within their starting region

If ISCs are intrinsically different in different subregions, then their progeny might only be able to differentiate into cells from that subregion. In contrast, if regional differences are induced by signals received from the gut lumen, muscle, enteric nerves or other sources, then ISCs near a border should on occasion generate daughter cells from both regions. We noticed that clones in the midgut sometimes run for a considerable distance perpendicular to the a/p axis (*Figure 5A*), but rarely if ever parallel to the axis. To investigate quantitatively whether there are restrictions on cell movement and/or differentiation along the a/p axis, we marked stem cells at a relatively low frequency to avoid generating two adjacent clones derived from different regions. Progeny of individual clones were then analyzed with respect to particular regional boundaries defined using appropriate GAL4 lines or other region-specific markers. Among more than 3000 sparse clones generated throughout the midgut, we identified rare 'boundary clones' that contained a single stem cell, at least four total cells and that contacted a marked boundary from one side or the other. Each boundary clone was carefully analyzed by confocal microscopy so that the location of its stem cell (marked by Delta), the number, cell type and location of the downstream cells (marked by lacZ and DAPI) and their precise relationship to the boundary in question (marked by GFP) could be determined (*Figure 5—figure supplement 1*). We then recorded clone size, the number of cells that contacted the boundary, and the number of cells, if any, that crossed the boundary and successfully differentiated into cells of the adjacent region as defined by regional marker expression (*Figure 5B,C*). It was difficult to obtain large numbers of valid boundary clones due to the low rate of clone induction, regional differences in stem cell activity, and the difficulty of analyzing boundary clones accurately using different regional markers. However, we characterized 49 unambiguous boundary clones contacting six different regional boundaries (*Table 1*).

The A1/A2 boundary was typical of most that were studied. Boundary clones that resulted from stem cell labeling within A1 (*Figure 5D,D'*) or A2 (*Figure 5E,E'*) contacted the border zone defined by R45D10::UAS-GFP expression, but never included cells from the opposite side. Either the clones did not cross the boundary, or they crossed but were unable to differentiate properly and rapidly died. Since the boundaries are not smooth or precisely perpendicular to the a/p axis at the cellular level, it is also possible that cells from an A1 stem cell that pushed into the anterior edge of the A2 zone simply caused a bulge in the boundary at that location (see *Figure 5D,D'*). Over time such cell movement might cause the boundary between the two regions to fluctuate slightly in shape as cells die and are replaced by cells from one side or the other. Similarly, clones failed to differentiate heterologously across four other studied boundaries (*Table 1*), including Fe/P1 (*Figure 5H,I*) and P1/P2 (*Figure 5J,K*). In P1/P2, we verified clonal behavior using more than one boundary marker (*Figure 5J,K*).

Clones readily crossed one boundary, between LFC and Fe, and differentiated or transformed into cells characteristic of the opposite side. We induced ISC clones and identified 17 boundary clones touching the LFC/Fe border, using Fe region-specific expression of Ferritin-GFP to mark the boundary. The behavior of clones at this boundary was very different than at A1/A2. 15 of 17 clones crossed the boundary, and did so readily from either direction (*Figure 5F,G*). This was not an artifact of Ferritin-GFP expression because crossing was also observed using R46B08 to mark the LFC/Fe boundary, whereas clones failed to cross from Fe into P1 using Ferritin-GFP. Thus, despite the regional differences in enterocyte morphology and gene expression described earlier, the stem cells within LFC and Fe can each generate at least some cells characteristic of the other region. We concluded that our method efficiently identified clones that produced cells from two different regions across a boundary when such behavior occurred.

Since the average size and border contact of boundary clones was similar in all the regions (*Table 1*), the behavior of clones in the other regions could be compared quantitatively to clones at the LFC/Fe boundary to determine if their failure to differentiate across a boundary was statistically significant. For example, 88% of boundary clones on the LFC/Fe boundary crossed and generated cells on the other side (N = 17). In comparison, 0% of boundary clones crossed the A1/A2 border (N = 6), 0% of boundary clones crossed the A3/Cu border (N = 3), 0% of boundary clones crossed the Cu/LFC border (N = 4), 0% of boundary clones crossed the Fe/P1 border (N = 7), and 0% of boundary clones crossed the P1/P2 border (N = 12). These differences are all statistically significant, since the probability that even three boundary clones would fail to cross a junction equivalent to the LFC/Fe boundary is only $(1–0.88)^3 = 0.0017$. To further test the significance of the results, we reasoned that there should be a

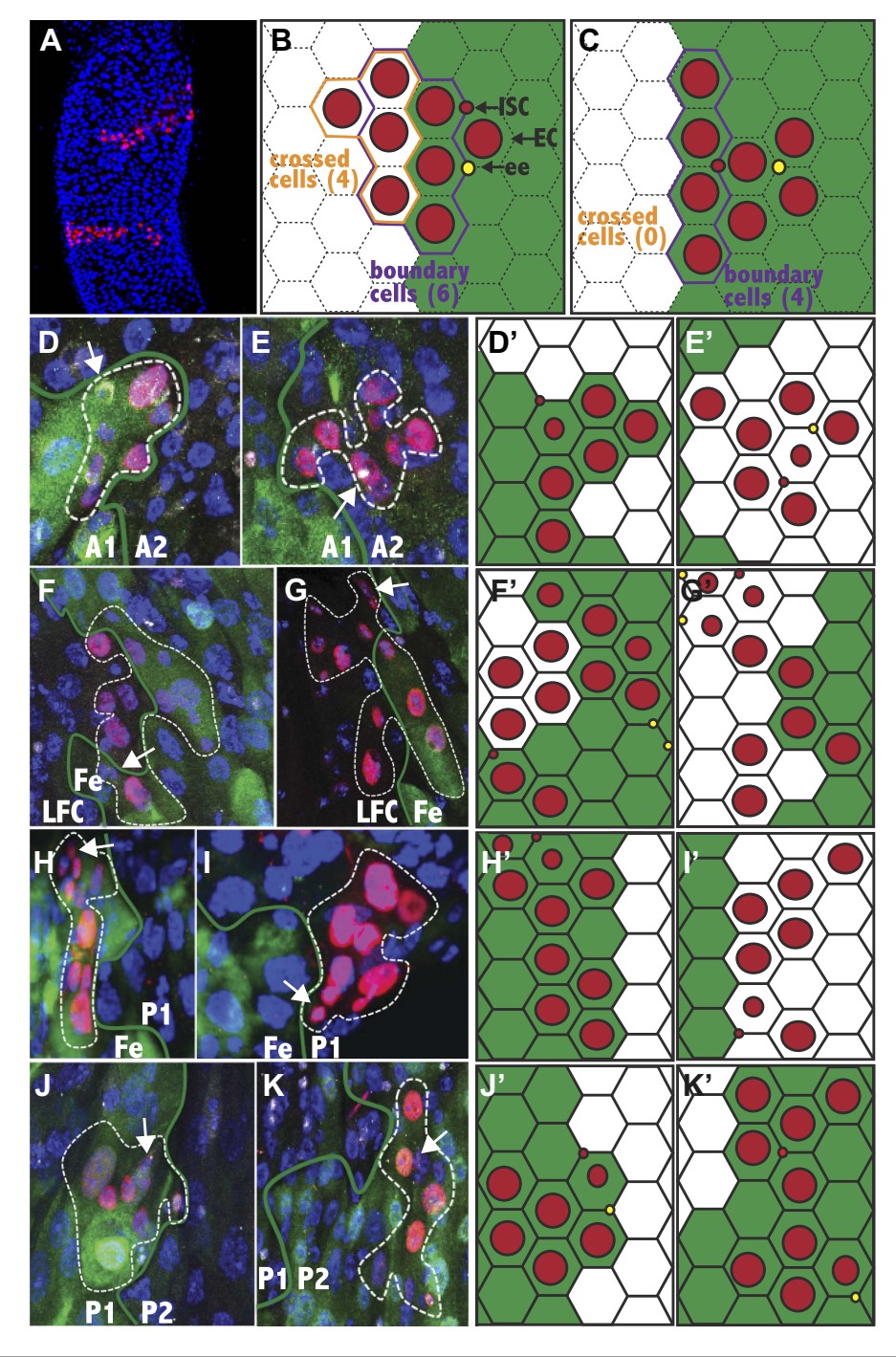

**Figure 5.** Stem cells are frequently compartmentalized. (**A**) Two clones from a 25dphs gut showing elongation perpendicular to the a/p axis. (**B–C**) Expected stem cell clonal (red circles) distribution near a regional boundary (green line) in the absence (**B**) or presence (**C**) of stem cell regional autonomy. (**B**) Regionally multipotent stem cells are predicted to produce progeny on both sides of the boundary. (**C**) Regionally autonomous stem cells generate clones that do not cross the boundary. The diagrams show how boundary clones were scored: boundary cells (purple outline) crossed cells (orange outline), enterocyte (large red circle), stem cell (small red circle), enteroendocrine cell (open red circle). (**D–K**) Fluorescence micrographs showing ISC clones (red), region specific EC expression (green), ISCs (white cytoplasmic), ee's (white nuclear), and DAPI (blue). Green lines: regional boundary determined by region-specific GFP expression (green). White dashed line: outline of clone. White arrow: Isc.
*Figure 5. Continued on next page*

*Figure 5. Continued*

(D and E) Green: 45D10-Gal4,UAS-GFP labels A1. (D) Clone originating and remaining in A1. (E) Clone originating and remaining in A2. (F and G) Green: 46B08-Gal4,UAS-GFP labels Fe. (F) Clone originating in the Fe region that crossed the LFC-Fe boundary. (G) Clone originating in the LFC region that crossed the LFC-Fe boundary. (H and I) Green: Ferritin-GFP labels Fe region. (H) Clone originating in and remaining in Fe. (I) Clone originating and remaining in P1. (J) Green: 46B08-Gal4,UAS-GFP labels P1. Clone originating and remaining in P1. (K) Green: 50A12-Gal4,UAS-GFP labels P2. Clone originating and remaining in P2. (D'–K') Schematic diagrams of clones in respect to regional boundaries extrapolated from (D–K).

The following figure supplements are available for figure 5:

**Figure supplement 1**. Analysis of boundary clones at single cell resolution.

correspondence between the number of boundary cells and the number of boundary-crossing cells, on average, in the absence of a constraint. In boundary clones at the LFC/Fe border, there were 0.63 crossing cells for every boundary cell (N = 115). In contrast, there were 0 crossing cells for 22 boundary cells at the A1/A2 border, 0 crossing cells for 14 boundary cells on the A3/Cu border, 0 crossing cells for 10 boundary cells on the Cu/LFC border, 0 crossing cells for 21 boundary cells on the Fe/P1 border, and 0 crossing cells for 53 boundary cells on the P1/P2 border. The absence of crossing cells differs significantly from expectation based on LFC/Fe (e.g., at the Cu/LFC border with ten boundary cells, $p=0.014$; $\chi^2 = 6.15$). However, the limited number of clones and boundary cells that could be obtained at the Cu/LFC and A3/Cu borders does limit our ability to detect cell crossing is inhibited in both directions. At the Cu/LFC border, Cu stem cell progeny cannot cross, but it is not possible to conclude that LFC progeny lack this capacity. The restriction at the A3/Cu border also is known to apply only from the A3 side. A summary of the compartmentalization of stem cell clones is shown in *Figure 6A*.

## Enteroendocrine tumor cells arise and progress differentially between regions

Disrupting Notch signaling in ISCs leads to enteroendocrine tumor formation. We investigated whether these tumors arise with the same frequency throughout the midgut by knocking down Notch signaling in ISCs and EBs using the esg-GAL4 driver, or by feeding flies the Notch inhibitor DAPT. Several days after reducing Notch activity, clusters of diploid tumor cells began to appear in the tissue. Surprisingly, tumorous cells arose and proliferated much more extensively in some midgut subregions than in others (*Figure 6B*). The P1 region consistently had the largest tumors and earliest onset, although tumors also arose rapidly in P3 and the posterior part of A2. In contrast, we never observed tumors within A1, even after long periods of exposure to these agents. As the tumor cells expanded they

**Table 1.** Clonal analysis of stem cell autonomy

| Region 1 (ISC) | Region 2 | Clones | Boundary clones | Clones crossed | Total cells | Boundary cells | Crossed cells | p value1 | Note |
|---|---|---|---|---|---|---|---|---|---|
| A1 | A2 | 52 | 2 | 0 | 21 | 10 | 0 | p<0.05 | |
| A2 | A1 | 250 | 4 | 0 | 34 | 12 | 0 | p<0.01 | |
| A3 | Cu | 86 | 3 | 0 | 29 | 14 | 0 | p<0.01 | |
| Cu | A3 | | 0 | 0 | 0 | 0 | 0 | n.a. | |
| Cu | LFC | 94 | 3 | 0 | 16 | 7 | 0 | p<0.05 | |
| LFC | Cu | 24 | 1 | 0 | 8 | 3 | 0 | n.s. | |
| LFC | Fe | 67 | 7 | 7 | 52 | 37 | 23 | | |
| Fe | LFC | 235 | 10 | 8 | 63 | 36 | 25 | | |
| Fe | P1 | 172 | 4 | 0 | 26 | 12 | 0 | p<0.01 | |
| P1 | Fe | 833 | 3 | 0 | 24 | 9 | 0 | p<0.01 | |
| P1 | P2 | 560 | 7 | 0 | 62 | 28 | 0 | p<0.01 | |
| P2 | P1 | 952 | 5 | 0 | 45 | 25 | 0 | p<0.01 | 2 dying |

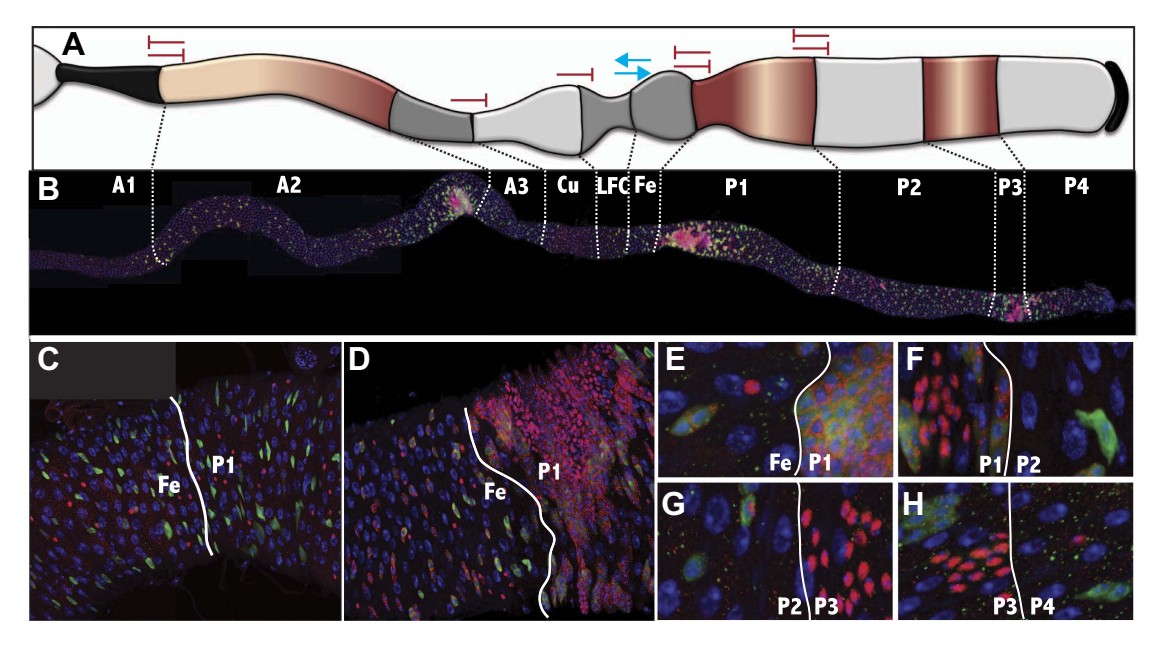

**Figure 6**. Midgut subregions differ in the production and movement of ISC/enteroendocrine tumor cells. (**A**) Summary of barriers to cross-regional stem cell differentiation. The results of testing whether stem cell clones are able to span six of the midgut regional boundaries are indicated graphically. Blue arrows indicate that stem cell progeny could cross the indicated boundary in the direction shown, and differentiate into regionally appropriate cells. Red inhibitor symbols indicate that clonal crossing and differentiation from the indicated directions were not observed. Maroon shading shows the regions of the midgut in which enteroendocrine tumors arise preferentially when Notch signaling is inhibited. (**B**) Low magnification view of the entire midgut from an animal expressing UAS-N[RNAi] driven by esg-GAL4 showing accumulation of tumor cells in regions A2, P1 and P3. (Green: UAS-GFP expression, Red cytoplasmic: Delta expression in ISCs, Red nuclear: Prospero expression in ee's, Blue: DAPI in all nuclei, White lines: regional boundaries). (**C**) Control expression in the Fe and P1 regions in the absence of UAS-N[RNAi]. (**D** and **E**) Higher magnification of a midgut as in (**B**) ee and ISC-like cells are evident in P1 but not in Fe; (**F–H**) tumor cells induced by esg-GAL4 driven expression of Notch[DN] appear to respect regional borders between P1/P2 (**F**), P2/P3 (**G**), and P3/P4 (**H**). Markers are as in (**B–E**).

appeared to respect the cellular boundaries with adjacent midgut regions. For example, the anterior border of P1 tumor cells corresponded closely with the Fe/P1 boundary based on counts of cell diameters from the Cu region (*Figure 6D–E*). The posterior boundary of P1 tumor expansion mapped near or at the P1/P2 border (*Figure 6F*), while the anterior and posterior boundaries of P3 tumors also corresponded with their normal junctions with P2 (*Figure 6G*) and P3/P4 (*Figure 6H*). The same regional specificity was observed when Notch signaling was inhibited using N[RNAi], treatment with the Notch inhibitor DAPT, or by expressing a Notch-dominative negative allele N[DN]. These results indicate that regional epigenetic differences have a large effect on the development of midgut enteroendocrine tumors and that regional boundaries restrict tumor cell movement into adjacent tissue compartments.

## Discussion

### The Drosophila midgut differs regionally along its length

Our experiments significantly expand previous knowledge of regional variation within the Drosophila midgut. At the levels of cell morphology, cell behavior and gene expression, the midgut is much more highly organized than a uniform cellular tube containing an acidic middle region or 'stomach'. Regionalization likely supports the complex metabolic tasks carried out by the midgut. Ingested food goes through multiple intermediate stages during digestion and these steps may be most efficient if carried out in a controlled sequence (*Figure 3*). Some of these steps are associated with an array of bacterial species that constitute the normal intestinal microbiome (*Ryu et al., 2008*; *Shin et al., 2011*). Our work will allow the function of genes, pathways, cells and regions within the midgut to be tested in digestion, tissue maintenance, microbiome function and immunity.

After this work was submitted for publication, *Buchon et al. (2013)* also described midgut regionalization; they classified five major midgut divisions (R1–R5), 13 total subregions (e.g., R1a and R1b), and 2 microregions ($B_{R2-3}$, $B_{R3-4}$). Converting our counts of cell number within the 10 regions (*Figure 1C*) to the fractional length coordinates used by *Buchon et al. (2013)* suggests a close correspondence between the studies: A1 = R1a + R1b; A2 = R2a + R2b; A3 = R2c; Cu = R3a + R3b; LFC = R3c; Fe = $B_{R3-R4}$, P1 = R4a + R4b; P2 = R4c; P3 = R5a; P4 = R5b. In the case of single regions in our system that Buchon et al. split in two, we had also usually noted differences in gene expression; for example, between the anterior and posterior Cu region (*Figure 1D*, *Figure 1—figure supplement 2*; *Strand and Micchelli, 2011*; *Buchon et al., 2013*) or the subregions of A2 and P1 that differ in lipid accumulation. We emphasized enterocyte morphology and qualitative gene expression differences in defining subregions. In contrast to our study, *Buchon et al. (2013)* view constrictions to be of primary importance and define two constrictions as micro-regions in their own right. We found that enterocyte subregions abut directly, and favor the view that constrictions are mesodermal features, not microregions.

## Differences in ISCs contribute to Drosophila midgut regionalization

Our results argue strongly that the striking regionalization of structure and gene expression within the midgut is maintained at least in part by regional differences between their resident stem cells. In the midgut subregions surrounding five different boundaries, we did not detect a single stem cell that produced differentiated cells on the opposite side of the boundary, that is from a region different from the one in which it resided. All the 'non-crossing' clones contacted the regional boundary, and in 78% the founder stem cell was located on or within one cell of the boundary, such that progeny cells could have reached the adjacent region prior to differentiation. In contrast, boundary clones with the same general properties almost always crossed the LFC/Fe boundary, showing that 'non-crossing' behavior would not occur by chance. Consistent with the existence of epigenetic differences in stem cells that limit trans-regional differentiation, clones frequently pushed into adjacent regions at the boundary, but retained their autonomous identity based on marker expression. Mechanical and/or adhesive forces may also contribute to maintaining some regional boundaries. Indeed, the tendency of tumor cells to respect regional boundaries suggests that cell–cell interactions at boundaries are likely to be important as well as stem cell programming.

The behavior of ISC clones at regional boundaries is reminiscent of the behavior of clones in developing imaginal discs at 'compartment' boundaries (reviewed in *Dahmann et al., 2011*). In the developing Drosophila embryo and imaginal discs, the *engrailed* gene and *hedgehog* signaling play important roles in defining posterior compartments (*Dahmann and Basler, 2000*). We did not detect expression of *engrailed* or the closely related gene *invected* in any midgut region, and expression of *hedgehog* pathway components was similar on both sides of non-crossing boundaries such as Fe/P1. Dorsal ventral compartments in the developing wing are mediated by *apterous* and by Notch signaling (*Milan and Cohen, 2003*). *Apterous* was expressed only at very low levels throughout all regions and *Serrate* was found at significant levels only in posterior regions 2–4 where the gene is dispensable for cell differentiation (*Ohlstein and Spradling, 2007*). In the developing vertebrate brain, Hox genes are important in specifying developmental compartments (*Kiecker and Lumsden, 2005*). However, Hox genes are only expressed at very low levels in endodermal cells during embryogenesis (*Hoppler and Beinz, 1994*) and these genes were very weakly expressed in our RNAseq studies, perhaps due to expression in non-endodermal cells within these samples. Consequently, the genetic basis for adult midgut compartmentalization probably differs from previously studied examples of tissue regionalization.

The homeotic transcription factor *labial (lab)* is an outstanding candidate for a regional regulatory factor. In the embryonic and larval gut, *lab* is required for Cu cell specification, differentiation and maintenance (*Hoppler and Beinz, 1994*; *Dubreuil et al., 2001*). The gene is expressed in copper cells, but not elsewhere in the larval midgut, and we observed similar specificity of *lab* expression in the adult. When *lab* is mis-expressed during embryonic development in other midgut regions, the copper region can expand (*Hoppler and Bienz, 1994*). Endodermal cell identity along the a/p axis may be determined by signals from adjacent mesoderm during embryogenesis (*Bilder and Scott, 1998*), and then fixed by the induction of secondary factors such as *lab*. Gene expression within the midgut muscles might play a similar role in the adult midgut, however, expression boundaries of muscle genes were frequently offset with respect to endodermal regions (*Figure 1E*). Whether this bears any relationship to the documented offset in homeotic gene expression between the ectoderm and

visceral endoderm in embryonic development (*Lawrence and Morata, 1994*) remains unclear. A key question is whether individual or combinations of differentiation regulators analogous to *lab* specify other midgut subregions in which the ISCs fail to generate cells across regional boundaries. The RNAseq data should provide a valuable resource in identifying such factors. For example, one potential candidate, the homeotic gene *defective proventriculus (dve)*, functions in copper cells (*Nakagawa et al., 2011*) and its expression was observed to fall sixfold between Fe and P1.

## Origin of regional differences within the midgut

One remaining question is whether a pre-existing pattern of larval midgut subdivision plays any role in the origin of adult midgut organization. The larval gut has a middle acidic region containing copper cells and an iron region like the adult tissue (*Poulson and Bowen, 1952*), and EM studies show additional morphological differences (*Shanbhag and Tripathi, 2009*). However, it is not known whether regions analogous to other eight midgut domains described here exist in larvae. The larval midgut contains nests of diploid intestinal precursors that proliferate following pupariation to build the adult gut and establish its ISCs (*Jiang and Edgar, 2009*; *Mathur et al., 2010*). Larval midgut domains might serve as a template for adult regionalization if gut precursor cells within each region already differ autonomously and do not mix during pupal development. However, cells do cross boundaries between the hindgut and midgut during pupal gut development (*Takashima et al., 2013b*). Identical regionalization within larval and adult guts might be disadvantageous to species with very different larval and adult diets, hence many adult midgut regions are likely to be established de novo or to be re-specified during pupal development.

## Stem cell regionalization may be widespread

Many mammalian tissues such as skin, muscle, lung, liver, and intestine contain thousands of spatially dispersed stem cells, like the Drosophila midgut. Our studies raise the question of whether these tissues exhibit finer grained regional patterns of gene expression than has been previously recognized, patterns that might be supported by small autonomous differences in their stem cells. Currently, the strongest indication for such regionalization comes from studies of the intestine. Lineage labeling shows that similar stem cells expressing Lgr5 exist along the mammalian gut (*Barker et al., 2007*) despite the fact that enterocytes, enteroendocrine cells and bacterial symbionts differ regionally (*Rindi et al., 2004*; *Bradley et al., 2011*). For example, iron absorption in mammals takes place primarily in the duodenum (*Fuqua et al., 2012*), a specialized subregion of the small intestine located just downstream from the acidic stomach. This is similar to the position of the midgut iron region just downstream from the acid-producing parietal cells of the Cu and LFC regions. The antibacterial lectin RegIIIγ, which like Drosophila PGRPs recognizes bacterial peptidoglycans, is expressed most prominently in the distal region of the small intestine (*Cash et al., 2006*). The existence of tissue and stem cell regionalization in other mammalian tissues deserves further detailed investigation.

## Regionalization and cancer

The human large intestine is much more prone to cancer than the small intestine. Our studies suggest that regional differences in the properties of apparently similar stem cells and tissue cells contribute to such differences. The midgut zones most favorable for the expansion of Notch-deficient cells showed pre-existing differences in Notch signaling within the early enterocyte lineage. Delta expression did not decrease shortly after ISC division, as in other regions, and Notch signaling persisted throughout enterocyte development (*Figure 4Q*). Curiously, same tumor-prone regions with persistent Notch signaling also were enriched in lipid droplets (*Figure 1*). At present it is not clear how the altered signaling, regional metabolic activity and tumor susceptibility are related. Additionally, regional differences in the microbiome, as suggested by our observation of domain-specific expression of PGRP proteins, may also influence the occurrence of cancer. Gastric bacteria such as *Helicobacter pylori* contribute to stomach cancer (*Uemura et al., 2001*), while colonic *Bacteroides fragilis* likely promote gut cell DNA damage and colon cancer (*Wu et al., 2009*). Regional tissue differences likely also affect rates of tumor progression and metastasis. These observations emphasize the importance of understanding tissues region by region.

In sum, the Drosophila midgut provides an outstanding tissue in which to explore and understand the significance of intrinsic stem cell differences. We identified GAL4 drivers that allow gene expression to be manipulated in all intestinal cell types, including cells such as circular muscle and enteric neurons

that are thought to contribute to niche function. Will altering the expression of genes that normally differ between regions cause ISCs to generate cells with heterotypic characteristics? Such studies might eventually make it possible to stimulate medically useful responses from the endogenous stem cells that remain within a diseased tissue.

## Materials and methods

### Strains and crosses

Strains of *Drosophila melanogaster* were obtained from Bloomington Stock Center (http://flystocks. bio.indiana.edu/) unless otherwise indicated. Genetic symbols are described in Flybase (*McQuilton et al., 2011*). Flies for all experiments were maintained either at a constant 25°C or between 23–25°C with 15–25 flies per vial and tossed onto new food every 2–3 days to ensure that midguts behaved in a reproducible manner that was minimally influenced by external variables. Janelia lines (*Jenett et al., 2012*) were obtained from JFRA; further information about each line is available online (www.janelia.org/gal4-gen1). Sources of other strains were: esg-GAL4;tub-GAL80ts,UAS-GFP (B Edgar); UAS-NotchRNAi (S Bray); UAS-Notch Dominant Negative (UAS-N$^{DN}$) (I Rebay), and SREBP-reporter line (R Rawson; *Matthews et al., 2009*).

### Immunofluorescent staining

We dissected the gut tissue in Grace's buffer and fixed on a nutator shaker for 1 hr at room temp (or 16–20 hr at 4°C for Delta staining) in 4% paraformaldehyde (Electron Microscopy Sciences, Hatfield, PA; 32% paraformaldehyde, EM grade, cat:15,714) and 2% antibody wash (0.3%TritonX and 0.3% of 30%BSA in 1XPBS) in Grace's buffer. After fixation, guts were washed in antibody wash with continuous shaking three times for at least 20 min followed by primary and secondary antibody application for at least 12 hr each at 4°C with at least 12 hr of washing with antibody wash in between. After the secondary antibody incubation, samples were washed in antibody wash three times for at least 20 min. Finally, nuclei were stained with DAPI (1:5000 of Sigma, St. Louis, MO; cat:D 9542) diluted in 1XPBS for 30 min with shaking. Primary antibodies: mouse-α-Prospero (1:20), mouse-α-Delta (1:100), and mouse-α-Cut (1:20) from Developmental Studies Hybridoma Bank (http://dshb.biology.uiowa. edu/), chick-α-βGal from Abcam (Cambridge, MA; ab9361); and rabbit-α-GFP (1:1000) from Invitrogen (cat:A11122). Secondary antibodies from Invitrogen (Burlingame, CA) all used at 1:500: goat-α-mouse AlexaFluor488 (A11001), goat-α-mouse AlexaFluor568 (A11004), goat-α-mouse AlexaFluor633 (A21052), goat-α-rabbit AlexaFluor488 (A11034), goat-α-chick AlexaFluor568 (A11041). Nile red (72,485; Sigma) and Prussian blue (03,899; Sigma) were applied at the secondary antibody wash step, and utilized at 1:5000. Preparations were mounted in Vectashield (Vector Labs Inc. cat:H-1000), and examined with a LeicaSP5 confocal microscope.

### GAL4 line screening

931 lines from the Janelia GAL4 enhancer trap collection were crossed to the UAS-RFP, UAS-FLP, Ubi$^{p63}$–FRT-stop-FRT-nEGFP/CyO (*Evans et al., 2009*) and guts were dissected from at least three progeny females aged 3–7 days from each strain. Guts were fixed, stained to reveal GFP, Prospero and Delta, and examined in a confocal microscope. We found 63 lines with expression in the adult proventriculus, midgut epithelium, midgut muscle, and/or enteric nervous system. The expression of these lines were analyzed further by crossing to UAS-GFP.S65T (Bloomington 1522) and mated female flies were reared at 25°C with 15–25 flies per vial. Flies were transferred onto new food every 2–3 days and fed with dry yeast 2 days prior to dissection (between 6–10 days post eclosion). At least 10 guts were analyzed from each line to document expression. *Figure 1—figure supplement 2* presents the spatial pattern and cell types expressing GFP in those lines with midgut expression.

### Electron microscopy

Eight strains with region-specific GFP expression, R50A12-GAL4, R45D10-GAL4, R46B08-GAL4, R42C06-GAL4, R43D03-GAL4, R47G08-GAL4, R40B12-GAL4 and Ferritin-GFP (*Morin et al., 2001*), were crossed to UAS-GFP.S65T (Bloomington 1522) and progeny flies were reared at 25°C for 6–10 days. Mated females fed on dry yeast for 2 days prior to collection were dissected in Grace's buffer and appropriate midgut regions were isolated under a fluorescent dissecting microscope. At least five samples of each region were fixed for 1 hr at 4°C in 1% gluteraldehyde, 1% OsO$_4$, 0.1 M

cacodylate buffer, 2 mM Ca (pH 7.5). Following washing in cacodylate buffer (3X × 5 min) tissue was embedded in agarose at 55°C, rinsed for 5 min in 0.05 M maleate (pH 6.5), and stained for 1.5 hr in 0.5% uranyl acetate, 0.05 M maleate pH 6.5. After rinsing in H₂O, samples were dehydrated in an ethanol series (35% twice for 5 min, 50% for 10 min, 75% for 10 min, 95% for 10 min and 100% three times for 10 min), incubated in propylene oxide (2X × 10 min) and then in 1:1 propylene oxide:resin (Epon 812:Quetol 651 [2:1]), 1% silicone 200, 2% BDMA for 1 hr. After three changes of resin (1 hr each), resin was allowed to polymerize overnight at 50°C and then at 70°C overnight. Images were captured with a Phillips Tecnai 12 microscope and recorded with a GATAN multiscan CCD camera using Digital Micrograph software. We analyzed at least three longitudinal sections from each of the 10 proposed regions.

## Analysis of regional ISC parameters

The strains Hs-FLP;X15-29 and w;X15-33 that in combination allow lineage marking (see *Ohlstein and Spradling, 2006*) were crossed to UAS-GFP and a regional-GAL4 line, respectively, and their F1 progeny were intercrossed to generate the starting strains used. For regional proliferation rate studies, mated females 4 days post eclosion with the following genotype: Hs-FLP,UAS-GFP;X15-33/X15-29; regional-GAL4, UAS-GFP/+ or Hs-FLP;X15-33/X15-29; Fer-GFP/+, were heat shocked for 30 min at 37°C and dissected at multiple times thereafter. In each case, midguts were stained to reveal lacZ (clonal marker), Dl (to identify ISCs) and Prospero (to identify enteroendocrine cells), and the size and cellular content of ISC clones was analyzed.

## Analysis of regional stem cell autonomy

The ability of stem cell progeny to differentiate into cells from an adjacent subregion ('boundary crossing') was analyzed using the same flies and experimental design, except that flies were heat shocked for 10 min at 37°C (to ensure the clonal induction rate was low enough such that even large clones rarely touched nearby clones) and dissected at multiple time points up to 25 days after heat shock. Regional boundaries were marked with region-specific protein trap lines, GAL4-driver lines, or antibody stains. The A1/A2 boundary was indicated by R45D10 (A1 expression), A3/Cu by anti-Cut antibody or R50A12 (Cu parietal cell expression), Cu/LFC by anti-Cut antibody or R50A12 (Cu parietal cell expression), LFC/Fe by Ferritin-GFP or R46B08 (Fe expression), P1/P2 by R46B08 (P1 expression) or R50A12 (P2 expression), and P2/P3 by R50A12 (P2 expression). Clones containing a single ISC, at least four total cells, and contained at least one cell that contacted the boundary under study were termed 'boundary clones' and were analyzed further to determine the number and location of all cells relative to the boundary. A 'boundary cell' is any cell in the clone that neighbors midgut cell in a different subregion. A 'crossed cell' is any cell in the clone that is located past the boundary and that displays the marker expression of a midgut subregion different from that of its founder stem cell. A crossed clone is a clone that contains at least one crossed cell. Several classes of potentially ambiguous clones were not included in our tabulation to ensure accuracy. Clones lacking a single, clearly stained stem cell were excluded, as were clones that could not be accurately scored because they wrapped around the periphery of the squashed tissue. Additionally, clones touching regions where boundary marker expression could not be clearly scored were excluded. For each boundary clone, the total size, number of boundary-touching cells, number of crossed cells, if any, and the location of the ISC with respect to the border, was recorded.

## Analysis of Notch tumors

To perturb the Notch pathway we crossed UAS-N[RNAi}, and UAS-N[DN] to yw;esg-Gal4/CyO;tub-Gal80[ts],UAS-GFP/TM3 at 18°C to restrict transgene expression during larval and pupal stages. UAS-N[RNAi];esg-GAL4/+;tubulin-GAL80[ts],UAS-GFP/+, yw;esg-GAL4/UAS-N[act];tubulin-GAL80[ts],UAS-GFP/+, and yw;esg-GAL4/UAS-N[DN];tub-GAL80[ts],UAS-GFP/+ flies were moved to 29°C (permissive temperature for GAL4-mediated knockdown) upon eclosion, and dissected at multiple time points (no later than 16 days). Boundary limits were determined by counting cell diameters from the cut[+] copper cells, and by gut morphology.

## Isolation of gut regions for RNAseq

We identified GAL4 lines with regional expression, then dissected and sectioned gut regions based on the presence or absence of GFP under a fluorescent dissecting microscope. Flies were

maintained at 22–25°C for 6–8 days post eclosion, and given dry yeast for the 2 days prior to dissection. Gut regions were isolated, no more than five at a time, and moved to iced Tripure reagent (Roche/Boehringer Mannheim cat: 11667157001) to avoid RNA degradation (which occurred unless guts were processed within 30 min of isolation (or 10–15 min in the case of the copper region samples). After 25 or 50 gut regions were collected in 200 µl Tripure, they were homogenized. 600 µl of fresh Tripure was added, mixed, and allowed to stand at room temperature for 5–10 min. After adding 180 µl chloroform, samples were vortexed 2 × 45 s, and incubated at room temperature for 10 min. Samples were then centrifuged for 15 min at 12,000 rpm at 4°C, and the aqueous layer was moved to a fresh tube. RNA was precipitated by adding 400 µl of isopropanol, vortexing for 15 s, and incubating at room temperature for 15 min, and centrifuging at 12,000 rpm for 15 min at 4°C. After washing the pellet with 75% EtOH, the RNA was dried in air dry for 5 min, re-suspended in 50 µl nuclease free $H_2O$, and stored at −80°C. cDNA libraries were constructed from poly(A)-selected RNA using Illumina TruSeq RNA Library Prep Kit v2, and sequenced using a HiSeq2000.

### Analysis of RNAseq data

Fpkm values for genes in all samples were calculated using Bowtie2 v 2.0.6, TopHat v 2.0.7, and Cufflinks version 2.02, with Refseq annotation file dm3. Triplicate samples independently prepared and analyzed from each region gave fpkm values between 50,000 and 0.1 for about 10,000 of 15,600 annotated features. Expression values were highly reproducible between replicates ($R^2 > 0.95$), except for a small subclass of outlier genes, including many RNA genes. Genes in which the fpkm standard deviation (SD)/mean >1 within the replicates were excluded, since analyses of reads suggested that the divergence in such cases was usually due to inconsistent read alignment in the presence of nearby or overlapping genes. A small subset of genes showed possible physiological variation between replicates. Gene ontology analysis based on average fpkm was carried out for differentially expressed and for highly expressed gene sets within each subregion (*Supplementary file 1B*) using DAVID software (*Huang et al., 2008a*, *2008b*). These data are available at the NIH Geo Website under accession GSE47780.

## Acknowledgements

We are grateful to Todd Laverty, Gerry Rubin and the Janelia Farm Research Institute for providing access and assistance in screening lines from the Janelia Farm GAL4 collection. We thank all the lab members who helped to dissect, stain and screen these lines (Joan Palupa, Will Yarosh, Vicki Losick, Jianjun Sun, Don Fox). In addition, we acknowledge Vicki Losick for endless advice, clarification, interest, and encouragement, Matt Sieber for invaluable insights in analyzing the data, Mike Sepanski for the highest quality electron microscopy, and other members of the Spradling lab for support and comments on the manuscript.

## Additional information

### Funding

| Funder | Author |
| --- | --- |
| Howard Hughes Medical Institute | Allan C Spradling |

The funder had no role in study design, data collection and interpretation, or the decision to submit the work for publication.

### Author contributions

AM, Conception and design, Acquisition of data, Analysis and interpretation of data, Drafting or revising the article; ACS, Conception and design, Analysis and interpretation of data, Drafting or revising the article

## Additional files

### Supplementary files

• Supplementary file 1. (**A**) RNASeq analysis of Drosophila midgut. (**B**) Regional expression of midgut genes.

## Major dataset

The following dataset was generated:

| Author(s) | Year | Dataset title | Dataset ID and/or URL | Database, license, and accessibility information |
|---|---|---|---|---|
| Marianes A, Spradling A | 2013 | Physiological and stem cell compartmentalization within the adult Drosophila midgut | GSE47780; http://www.ncbi.nlm.nih.gov/geo/query/acc.cgi?acc=GSE47780 | Publicly available at GEO (http://www.ncbi.nlm.nih.gov/geo/). |

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
