## [Decision Letter]

Thank you for sending your work entitled “Physiological and stem cell compartmentalization within the Drosophila midgut” for consideration at *eLife*. Your article has been favorably evaluated by a Senior editor and 3 reviewers, one of whom is a member of our Board of Reviewing Editors. One of the reviewers, Bruce Edgar, has agreed to reveal his identity.

The Reviewing editor and the other reviewers discussed their comments before we reached this decision, and the Reviewing editor has assembled the following comments to help you prepare a revised submission.

The manuscript describes an extensive and careful characterization of regionalization of the adult Drosophila gut. The authors identify ten sub-regions based on differential gene expression, combining the expression patterns of GAL4 driver lines and RNA seq. of dissected segments of the gut. The data on regionalized expression is extensive and of good quality, and will be essential for future functional studies of regionalized gut functions in these different cell subtypes. The authors investigate whether the stem cells in all sub-regions are equivalent and demonstrate apparent lineage restricted compartment boundaries. They include some clonal analysis that reveals, potentially, the presence of clonal restriction boundaries that partition different subsets of stem cells. They observe that different regions of the gut are more susceptible to tumorigenesis upon inhibition of Notch signalling, suggesting regional differences in stem cell behavior. Although the authors have not addressed the mechanisms underlying these differences, documenting them is an important first step towards elucidating the mechanisms that control differential stem cell activity. Overall this is an impressive amount of work and will be useful for researchers working with this interesting system. It overlaps somewhat with recent “atlas” type papers from Tripathi and Lemaitre, but has unique aspects and different gene expression and Gal4 databases, so it will serve to complement these other studies.

Major comments:

1) The suggestion of regional compartment boundaries within the midgut, derived from clonal analysis, is one of the paper's more interesting discoveries. However the clonal data supporting this is a very small data set and cannot be considered comprehensive or even conclusive for a few boundaries. Far too few clones were assayed, and the data provided is just text, without tables, a proper statistical treatment, or a map (text and a figure are provided). This aspect of the paper could be vastly improved if hundreds of clones were assayed, a statistical treatment was applied, and a map was provided. As is, the data are merely suggestive of an interesting phenomenon.

2) In the Introduction the authors make the intriguing suggestion that intestinal stem cells may “differ regionally in their ability to replace damaged cells and to spawn tumors”. However, there are no experiments on damage and replacement. Does susceptibility to tumors necessarily reflect the ability to replace cells after damage?

3) The graph of ISC cell cycle times in different regions is probably not an accurate assessment of normal ISC division times. This is because the times were derived by clonal analysis using a heat shock-flip, and heat shock causes a long-term increase in ISC division rates in the adult midgut. A number of labs have confirmed this effect of heat shock. This issue has confounded earlier conclusions regarding timing of ISC cell cycles. The caveat needs at least to be mentioned here.

4) The authors should cite the recent similar paper published in Cell Reports from Buchon, et al. (2013), and comment on the differences in the regional map provided in that paper with theirs. If possible, this new paper should provide a revised map that covers both papers. The differences in the two maps are small and should be reconcilable.

5) The paper provides a very useful resource to labs studying Drosophila intestinal development and physiology. More data to support the concept of compartment boundaries and the contention that there are regional differences in the ability to replace damaged cells would broaden its interest.

---

## [Author Response]

*1) The suggestion of regional compartment boundaries within the midgut, derived from clonal analysis, is one of the paper's more interesting discoveries. However the clonal data supporting this is a very small data set and cannot be considered comprehensive or even conclusive for a few boundaries. Far too few clones were assayed, and the data provided is just text, without tables, a proper statistical treatment, or a map (text and a figure are provided). This aspect of the paper could be vastly improved if hundreds of clones were assayed, a statistical treatment was applied, and a map was provided. As is, the data are merely suggestive of an interesting phenomenon*.

The original clonal data comprised more than 1200 clones, of which 24 contacted one of two different regional boundaries. The conclusions drawn, though limited to these two boundaries, were valid and statistically sound. However, we agree with the reviewers’ main points that these data were not presented in sufficient detail, and moreover that this aspect of the work should be expanded to encompass more boundaries. We have now addressed these issues. From more than 2000 additional clones we have characterized 25 more boundary clones contacting 4 additional boundaries. Interestingly, the clones do not cross any of these new boundaries. So, instead of one crossable boundary and one non-crossable boundary, we now show that clonal crossing is not observed at 5 of 6 boundaries. Thus, compartmentalization is the rule, not the exception, further enhancing the interest of these experiments.

We revised the manuscript to make the clonal experiments easier to follow and analyze in detail, as requested. The experiments now have their own section in the Materials and methods section, detailing the genotypes needed to separately score both clones and regional boundaries. They now have their own figure, Figure 5, which contains two new panels explaining exactly how clones were scored (Figure 5). We added a new Table 1 that tabulates all the clonal data scored as described, including the number of boundary clones, boundary cells and cells crossing the boundary for every tested region. We show examples of clones from multiple regions and diagrams illustrating their interpretation (Figure 5). We added a figure (Figure 5–figure supplement 3) showing separated channels to document the accuracy of our clonal mapping relative to subregional borders. Finally, we added a new figure panel (Figure 6), summarizing the outcomes at the six borders analyzed.

Additionally we present a completely new and clearer statistical analysis showing that the compartmentalization results are significant at all the boundaries, relative to the LFC/Fe boundary. This is true despite the difficulty of obtaining large numbers of boundary clones. For example, 88% of boundary clones on the LFC/Fe boundary cross to the other side (N = 17). In comparison, 0% of boundary clones on the A1/A2 border cross (N = 6), 0% of clones on the A3/Cu border cross (N = 3), 0% of clones on the Cu/LFC border cross (N = 4), 0% of clones on the Fe/P1 border cross (N = 7), and 0% of clones on the P1/P2 border cross (N = 12). Since the large clones analyzed almost always cross at LFC/Fe, and were of similar size and border touching characteristics in all regions (Table 1), clonal behavior at the other boundaries is significantly different (even for N = 3, p=0.0017). Another way to demonstrate this significance is to note that for Cu/LFC, there are 0.63 crossing cells for every boundary cell (N = 115). In contrast, there are 0 crossing cells at A1/A2 border (N = 22), 0 crossing cells on the A3/Cu border (N = 14), 0 crossing cells on the Cu/LFC border (N = 10), 0 crossing cells on the Fe/P1 border (N = 21), and 0 crossing cells on the P1/P2 border (N = 53). All these numbers are significantly different from expectation based on LFC/Fe (e.g., even with N = 10 boundary cells, the chance of zero crossing cells is just p=0.014; χ^2^ = 6.15). However, the number of clones and boundary cells available does limit certain conclusions, and these are now spelled out in detail. For example, at the Cu/LFC border, it is not possible to conclude that the restriction applies in both directions, rather than just from Cu stem cells. The restriction at the A3/Cu border also is known to apply only from the A3 side. These limitations are indicated in the text and the diagram in Figure 6.

This new data solidifies and extends our previous conclusions about compartmentalization, as requested. It was not possible to exhaustively test every boundary because an appropriate GAL4 line demarking every boundary is not currently available, and scorable boundary clones arise at variable rates due to differences in stem cell activity and marker quality in the different subregions. However, we feel strongly that the clonal data presented, which represents a large amount of effort, establishes that in multiple regions of the midgut, cells are normally replaced only by the local ISCs from that region. This is further supported by the tumor data. It would be unreasonable to expect that the same paper that (co-)discovered the existence of midgut subregions, characterized them sufficiently to even contemplate such experiments, and established the existence of compartmentalization, would exhaustively analyze their clonal properties.

*2) In the Introduction the authors make the intriguing suggestion that intestinal stem cells may “differ regionally in their ability to replace damaged cells and to spawn tumors”. However, there are no experiments on damage and replacement. Does susceptibility to tumors necessarily reflect the ability to replace cells after damage*?

We were referring to the routine day-to-day activity of midgut stem cells that generate new cells in order to replace cells damaged by the stressful environment digestion produces, even in the absence of exogenous pathogens or toxins. We found that ISCs differ in their ability to carry out this routine replacement of damaged cells because in at least 7 of the 10 subregions our experiments show that ISCs are unable to replace cells in an adjacent region (Figure 6). The many regional differences in transcription factor expression that we uncovered may help explain this regional specificity, as has been documented already for labial and the Cu region. Despite the fact that the statement seems justified, we modified it to avoid any misunderstanding.

*3) The graph of ISC cell cycle times in different regions is probably not an accurate assessment of normal ISC division times. This is because the times were derived by clonal analysis using a heat shock-flip, and heat shock causes a long-term increase in ISC division rates in the adult midgut. A number of labs have confirmed this effect of heat shock. This issue has confounded earlier conclusions regarding timing of ISC cell cycles. The caveat needs at least to be mentioned here*.

Stresses that transiently induce the heat shock response, including brief elevated temperature, are part of the normal environment of Drosophila in the wild. Most of the claims that heat shock induces artifacts (e.g., [44]) are based on cell marking systems that involve the GAL4 system, which is itself affected by heat. The artifact may be the effect of heat shock on such marking systems rather than on the gut; e.g., a transient heat-induced increase in GAL4 activity could cause extra flip-out cell marking. Our experiments used a simple, reliable marking system that does not involve GAL4, has zero-background and is inactive in postmitotic cells; we applied only a single brief heat shock. Moreover, our interest here is primarily in documenting the *relative* differences in ISC activity between subregions, which were all treated the same. We now mention heat shock as an issue and so qualify our experiment. However, we believe that these data remain important and valuable as another way to document subregional differences.

*4) The authors should cite the recent similar paper published in Cell Reports from Buchon, et al. (2013), and comment on the differences in the regional map provided in that paper with theirs. If possible, this new paper should provide a revised map that covers both papers. The differences in the two maps are small and should be reconcilable*.

Our manuscript describes work carried out over a four-year period that was entirely independent of the work of Buchon, et al. (2013). The existence of their study became known to us only when they began to publicize their data at the same time we began presenting our studies in seminars; we both made poster presentations at the National Drosophila meeting. We did not examine their data in detail prior to preparing our manuscript and we submitted our paper prior to the publication of their *Cell Reports* paper. Credit for the discovery of detailed midgut regionalization deserves to be shared equally between the two groups.

We do agree that the existence of two independent co-discoveries of the highly regionalized nature of the midgut are valuable for the field. We are very positive about their work, which we view as complementary to our own, and we want to assist the field in “digesting” all this new information as efficiently as possible. Consequently, we have added a text section in the revised Discussion that briefly compares the two analyses of midgut regionalization. Based on our reading of their paper, the two studies are in fairly close but not complete agreement. We highlight the few differences, such as the nature and importance of the constrictions. Buchon, et al. (2013) present two of them as mini-regions in their own right lying between the other regions, whereas we locate the constrictions as features (caused mostly likely by overlying muscle) lying within or spanning regions whose cells directly contact each other. We sought to maintain the previously published regional names as much as possible, in order to facilitate access to the earlier literature, which goes back to the 1950s. Buchon, et al. (2013) renamed all regions based on the idea that the constrictions are the truly fundamental landmarks. In the end, the differences between the regions defined by the two groups are small, and workers in the field will decide which terms are most appropriate and useful.

*5) The paper provides a very useful resource to labs studying Drosophila intestinal development and physiology. More data to support the concept of compartment boundaries and the contention that there are regional differences in the ability to replace damaged cells would broaden its interest*.

Our paper goes beyond just describing regionalization, important as that is, and establishes a cellular mechanism–stem cell regional independence–that has great relevance and interest for all stem cell researchers, biologists, and doctors. Thus, similar-appearing stem cells in large tissues can no longer be assumed to be equivalent at the level of cell programming or gene expression, even though they make basically the “same” cell types. This new understanding will accelerate studies of stem cell biology, cell therapy, as well as work related to multiple aspects of gut physiology including the burgeoning field of the gut microbiome.

The work reported here appeals strongly to a wide variety of scientists, based on my experience presenting the data publically for the last seven months. It resonates with clinicians and mammalian researchers as strongly as any Drosophila work of which we are aware. Surgeons wonder whether similar stem cell regionalization might explain their own observations transplanting skin and blood vessels from one part of the human body into another. GI doctors are intrigued with the implications of this work for surgery on the human gut, which often has unexpected consequences. To further highlight the general significance of the principles revealed by our experiments, additional references were added describing data consistent with sub-regionalization in the mammalian small intestine. Some of these data suggest that even specific subregions (such as the iron region) might themselves be conserved between mammals and insects.

We are pleased the reviewers recognize that the manuscript also contains substantial new gene expression and genetic (GAL4 line) tools as a side benefit, and agree that these tools will facilitate future work on the midgut, along with complementary tools and data generated by the Buchon, et al. (2013). All the information and tools generated as part of this study are being shared. We have made all the expression analyses and the sequence data publically available via the NIH GEO site (accession number GSE47780). The GEO site is transferring the raw sequence data to the NIH short read archive, and linking it to the GEO summary. All the Janelia lines whose expression patterns we report are publicly available in the Bloomington stock center. We plan to send image files of the patterns to Janelia for inclusion on their GAL4 line website and have received verbal agreement that this can be arranged. No expression data were held back, including the rare lines expressing in subsets of ISCs.